# Effect of Organic Plant Ingredients on the Growth Performance of European Sea Bass (*Dicentrarchus labrax*): Nutritional Efficiency, Fillet Nutritional Indexes, Purchase Intention, and Economic Analysis

**DOI:** 10.3390/ani15162339

**Published:** 2025-08-10

**Authors:** Edilson Ronny Cusiyunca-Phoco, Manuel Saiz-García, Juan Benito Calanche-Morales, Ana Tomás-Vidal, Silvia Martínez-Llorens, Miguel Jover-Cerdá

**Affiliations:** 1Aquaculture and Biodiversity Research Group, Institute of Science and Animal Technology (ICTA), Universitat Politècnica de València, 46022 Valencia, Spain; saizgarciamanuel11@gmail.com (M.S.-G.); atomasv@dca.upv.es (A.T.-V.); silmarll@dca.upv.es (S.M.-L.); mjover@dca.upv.es (M.J.-C.); 2Faculty of Fisheries Engineering, José Faustino Sánchez Carrión National University, Huacho 15136, Peru; 3Department of Animal Production and Feed Science, Facultad de Veterinaria, Universidad de Zaragoza, 50009 Zaragoza, Spain; calanche@unizar.es

**Keywords:** organic vegetable proteins, sea bass, fillet nutritional indexes

## Abstract

This study demonstrates the use of organic plant ingredients with low levels of fishmeal inclusion in the diet of organic juvenile sea bass. The diets were formulated to contain 25%, 30%, and 35% fishmeal, with a fourth treatment comprising 30% fishmeal and conventional ingredients, serving as the control. The research demonstrated that the control diet resulted in optimal growth rates and biometric indices. However, specific variations were identified in the nutritional and qualitative characteristics of the fillet. However, these differences remained within the established reference ranges. Additionally, a higher level of preference and intention to purchase sea bass fed with organic feed was observed. It is evident that when offering an organic product, the types of ingredients utilized are of crucial importance. This is because they can facilitate consumer choice, thus achieving an encouraging outlook for organic aquaculture and promoting research.

## 1. Introduction

Sea bass (*Dicentrachus labrax* L.) is the second most important marine fish produced in the European Union aquaculture, with 90.883 tons, but the first in value, with 563 million euros. In Spain, sea bass is the most important fish, with 24.580 tons produced annually [1].

According to the report presented by APROMAR (2004) [2], in 2023, sea bass had a consumption share of 22.5 thousand tons in the last 12 months, ranking third after salmon and sea bream. The Scientific, Technical and Economic Committee for Fisheries (STECF) [3], in projecting a vision towards the year 2030, estimates a 20% increase in this sector, as did the Food and Agriculture Organization (FAO) [4] in 2022. Conversely, within the context of the Mediterranean region, the species that are most widely consumed are sea bass and sea bream. Collectively, these two species account for approximately 97% of the global production of both species, a factor that must be taken into consideration within the context of the global market [4,5,6].

When discussing the nutritional properties of sea bass (*Dicentrarchus labrax*), it is imperative to emphasize the significance of consumer preferences in Europe. This demographic highly values the species for its distinctive flavour, unique aroma, and low fat content, as it is classified as a semi-lean fish. These attributes contribute to the prevention of coronary heart disease and the enhancement of cognitive functions [7,8,9,10,11]. Conversely, the ongoing pursuit of sustainability in aquaculture has encouraged the investigation of innovative trends that consider alternative raw materials with a reduced ecological footprint, such as fishmeal and fish oil. These alternatives facilitate the optimization of circular economy strategies [12,13,14,15,16], thereby contributing to enhancements in zootechnical performance, nutritional efficiency, health, and technological progress associated with sea bass production [9,17,18].

Within aquaculture, the highest cost of production comes from the cost of feed, which comprises between 50 and 60% of total production; this percentage has fluctuated as aquaculture has developed in recent years. This underscores the vital importance of effective feed management, reliable access to ingredients for formulation, and the sustainability of feed production [17,19,20,21,22]. The composition of a sustainable feed with a reduced inclusion of fishmeal for carnivorous fish varies between 20% and 40%, depending on the proteins of vegetable origin. These can be readily substituted with soybean meal [23,24]. However, for sea bass (*Dicentrarchus labrax*), the level of inclusion without compromising their growth can be as low as 25% fishmeal [25], and for Japanese sea bass (*Lateolabrax japonicus*), 30% is considered [26]. In addition, other studies have demonstrated that increased utilization of soybean meal has been associated with diminished intake and reduced weight gain, in addition to deleterious effects on fish health, including alterations in intestinal integrity [24,26,27,28].

The European Sea bass (*Dicentrarchus labrax*) is among the most ecologically significant species in the Mediterranean region. The production of this species has increased substantially, with a rise from 2000 tons in 2015 to 2750 tons in 2020 [29,30]. Greece is positioned as the primary producer of this species within the European Union. Nevertheless, economic factors, including rising production costs and increasing retail prices, have given rise to concerns regarding investment in this sector, a sentiment that is particularly pronounced among both farmers and consumers. This circumstance impedes progress toward sustainable growth in organic sea bass production [31]. However, research is being conducted on species that are more in demand among European consumers, including sea bass, sea bream, and trout. Furthermore, efforts are being made to promote the production of these species, with the objective of achieving certification [32]. This approach entails the exploration of new markets and the establishment of a commitment to an ecological alternative, with a focus on fish welfare [16,33,34,35,36,37].

The utilization of organic inputs and by-products derived from organic agriculture and livestock has been demonstrated to hold considerable promise, given their status as a distinct market segment distinguished by an ecological and sustainable approach [31,38]. An increase in demand for these products has been observed, as well as a growing adaptation to innovative technologies in infrastructure [39,40,41,42,43]. However, restrictions imposed by the European Union, along with certain standards that this production system must meet [34,44,45,46], pose challenges for organic aquaculture in its search for sustainable protein sources for feeding carnivorous fish [16,35,36,37,40,47,48,49,50]. This context is essential for the development of an eco-friendly feed that does not compromise fish health. However, given the existence of reports indicating that there are no substantial disparities between organic and conventional aquaculture productions with regard to health parameters, such as blood tests and immunological responses, further research is necessary to ascertain the validity of these findings [50,51].

Given this problem, the feed industry has sought alternative protein sources, taking as an example those of vegetable origin that are more widely available on the market. However, such alternatives present implications due to the presence of anti-nutritional factors present, lower protein quality, and reduced palatability [52]. In recent years, the digestibility of plant-based proteins has improved through effective thermal processing—particularly extrusion during feed manufacture—and it is now possible to eliminate antinutrients, making the feed nutritionally suitable for fish in accordance with species-specific requirements [53,54,55,56,57].

Today, an increasing amount of information is available regarding diets containing organic inputs, as well as the relationship between conventional diets and organic diets, which is significantly influenced by Regulation (EU) No. 2018/848 [46]. According to Article 6, aquatic organisms may be fed with feed derived from sustainably exploited fisheries, in accordance with Regulation (EU) No. 1380/2013, or with organic feed composed of agricultural ingredients from organic production, including organic aquaculture, and natural, non-agricultural substances.

The approach of replacing fishmeal with other protein ingredients has been widely studied [15,24,58,59,60,61,62,63,64,65,66,67]. However, organic diets have not been extensively examined; only Tefal et al. [16] studied animal and plant ingredients derived from organic production in the absence of fishmeal. Their results, however, were inferior to those obtained with the control diet containing fishmeal. The issue of sustainability in organic aquaculture is of particular importance when formulating feed with reduced levels of fishmeal inclusion. The objective of the present study was to examine the impact of organic feeds varying in fishmeal content (25, 30, and 35%), combined with organic vegetable ingredients, as well as a control diet containing 30% fishmeal and conventional ingredients, on growth performance, biometric parameters, body composition, nutritional efficiency, fillet nutritional indexes, and economic viability.

## 2. Materials and Methods

### 2.1. Production System

The experimental study was conducted within a saltwater recirculation system, with a total volume of 65 m^3^. This system was equipped with a rotary mechanical filter and a gravity biofilter, each with an estimated capacity of 6 m^3^. The system also comprised twelve cylindrical fibreglass tanks, each with a capacity of 1750 L, divided into three tanks for each treatment. A system for the purpose of aeration was installed in all tanks located at the Aquaculture Laboratory of the Department of Animal Science of the Universitat Politècnica de València, Valencia, Spain.

### 2.2. Fish and Experimental Conditions

Organic sea bass (*Dicentrachus labrax*) fingerlings were obtained from the commercial hatchery Sonrionansa, located in the Autonomous Community of Cantabria in Spain.

At the beginning of the experiment, the fish had an average initial weight of 40 g, and 50 fish were distributed per tank. Fish were manually fed to satiety twice a day from Monday to Friday (at 9.00 a.m. and 2.00 p.m.) and once (10.00 a.m.) on Saturdays, over a 196-day period. The amount of ingested food was recorded daily. The temperature and oxygen dissolved in water (portable oximeter or OxyGuard Handy Polaris probe) were also checked daily, and salinity (refractometer, Hanna Instruments), pH (litmus paper strips), and concentration of ammonium, nitrites, and nitrates were monitored three times a week (MERCK test colourimeter). The average values obtained were as follows: 21.53 ± 2.2 °C (temperature), 7.62 ± 0.41 mg L^−1^ (dissolved oxygen), 7.02 ± 0.32 (pH), 16.15 ± 2.20 g L^−1^ salt (salinity 0/00), 0.18 ± 0.24 mg L^−1^ (ammonium), 0.25 mg L^−1^ (nitrites), and 92.32 mg L^−1^ (nitrates). The photoperiod used was natural, and all tanks had similar lighting conditions.

### 2.3. Experimental Feed

The experimental diets were manufactured at the Universitat Politècnica of Valencia’s facilities through a process of cooking-extrusion, using a semi-industrial extruder (CLEXTRAL BC 45, St. Etienne, France) belonging to the Department of Animal Science.

For the manufacture and formulation of experimental feed, the requirements established by organic production regulations were followed, using ecological raw materials which had their corresponding labelling and excluding formulation of non-permitted chemical substances such as synthetic amino acids. Four isolipid (17%) and isoproteic (46%) diets were formulated to evaluate the partial replacement of fishmeal with vegetable sources. Three organic experimental diets, FM 35 ECO, FM 30 ECO, and FM 25 ECO, containing 35, 30, and 25% fishmeal in their composition and complied with organic regulations. A fourth diet (FM 30 Control), also contained 30% fishmeal but included non-organic ingredients and served as the control.

Feed ingredients and composition of diets are shown in Table 1, Table 2 and Table 3.

### 2.4. Proximate Composition and Amino Acid Analysis

The dietary ingredients, feed (Table 1), and whole fish were analyzed according to the procedures established by the AOAC [68]. Dry matter (dried at 105 °C at constant weight), ashes (incinerated at 550 °C at constant weight), and crude protein were measured using a LECO CN628 machine, Geleen, The Netherlands, based on the Dumas method. Crude lipids were extracted using diethyl ether according to the ANKOM XT10 Technology Method 2 (2000).

The dietary composition and amino acid profile of the fish were examined utilizing a Waters high-performance liquid chromatography (HPLC) system, which includes two pumps (Model 515; Waters), an automatic sampler (Model 717; Waters), a fluorescence detector (Model 474; Waters), and a temperature control module, as outlined by Bosch et al. [69]: Subsequently, following the hydrolysis procedure, an internal standard of aminobutyric acid was incorporated into the analysis. The process of derivatization to facilitate the quantification of the amino acids employed 6-aminoquinolyl N-hydroxysuccinimidyl carbamate (AQC). The final results of the aforementioned analysis are displayed in Table 2.

### 2.5. Determination of Fatty Acid

Total fatty acid methyl esters (FAMEs) of lipids were produced directly, as indicated by O’Fallon et al. [70]. The samples were prepared according to the percentage of crude fat present in each sample; 12 samples were prepared with a weight corresponding to 20 milligrams of fat. 0.5 μL of the final sample was injected into a focusing A gas chromatograph (Thermo, Milan, Italy) equipped with a split/splitless injector and a flame ionization detector upon completing the procedures described in the aforementioned protocol. The separation of methyl esters was achieved by utilizing an SPTM 2560 fused silica capillary column (Supelco, PA, USA), which has 100 m × 0.25 mm × 0.2 μM film thickness. Helium was used as the transport gas, moving at 20 cm per second. The ratio for splitting was established at 1/100 while the sample was being injected. The split ratio was set at 1/100 during the injection of the sample. The initial oven temperature setting was 140 °C for five minutes, followed by an increase to 240 °C at a rate of 4 °C min^−1^ for 30 min. The cycle ended with a return to the original state. By comparing the retention times of specific fatty acids with Supelco’s fatty acid methyl ester criteria, one may identify them. Table 3 presents the results of the fatty acid composition of the feeds assessed in this study.

### 2.6. Growth and Nutrient Efficiency Indices

The following variables of growth and nutrient efficiency were determined as follows: specific growth rate (SGR), feed intake (FI), feed conversion rate (FCR), mortality (%). In order to obtain results that are more reliable, the biomass of dead fish reported in monthly reports was considered [35].

### 2.7. Biometric Indices

In order to obtain the biometric parameters, five randomly selected fish were euthanized from each tank, yielding a total of 15 fish per treatment in the final sampling. The sacrifice procedure was carried out by administering an overdose of anesthetic, whose formulation was based on clove oil (150 mg L^−1^). Subsequently, the total length (cm), total weight (g), carcass weight (g), liver weight (g), and visceral fat weight (g) were measured to calculate the biometric indexes: Condition Factor (CF), Viscerosomatic Index (VSI), Visceral Fat Index (VFI), and Hepatosomatic Index (HHI). Finally, following the appropriate identification and labelling of the fish samples, they were stored in a mixed manner at −30 °C for subsequent analysis.

### 2.8. Retention Efficiency

Protein, fat and energy retention were calculated as productive protein value (PPV), productive fat value (PFV), and productive energy value (PEV):PPV (%) = Retained protein (final fish protein × Final biomass (g)) × 100 − Initial fish protein × initial biomass (g)/Protein ingested (Kg of food ingested × % crude protein of feed)PFV (%) = Retained fat (final fish fat × Final biomass (g)) × 100 − initial fish fat × initial biomass (g)/Fat ingested (Kg food ingested × % crude fat of feed)PEV (%) = Retained Energy (Final fish energy × Final biomass (g)) × 100 − Initial fish energy × initial biomass (g)/energy ingested (Kg feed ingested × % feed energy)

Amino acid retention efficiency (AARE) were estimated as:AARE (%) = 100 × fish amino acid gain (g)/ingested amino acid (g).

### 2.9. Free Fillet Amino Acids

The free amino acids profile was carried out according to the protocol adapted from Bidlingmeyer et al. [71]. The free amino acid content of the fillet was evaluated by liquid chromatography (HPLC). A 2 g sample of fish meat was weighed, 10 mL of 0.1 N HCl was added, and the mixture was homogenized with Ultraturrax. Following homogenization, the sample underwent a series of centrifugation and filtration steps. Specifically, it was subjected to a 20 min centrifugation at 10,000× *g*, resulting in the interphase collection. This interphase was then filtered through a 0.45-micron PVDF filter, further purifying the sample. Subsequently, 300 μL of the filtrate was mixed with 50 μL of internal standard and 875 μL of acetonitrile. The mixture was left to stand until the following day. Subsequently, the sample underwent cooling and centrifugation at 10,000× *g* for five minutes. Subsequently, 300 μL of the resulting filtrate was subjected to centrifugal evaporation at 1750 rpm, 42 °C, and 35 min under vacuum conditions. Subsequently, 20 μL of drying reagent—comprising 3.5 mL of methanol, 0.5 mL of water, 0.5 mL of trimethylamine, and 0.5 mL of phenyl isothiocyanate was added, the mixture was shaken, and the solution was evaporated again until dry for 20 min. Subsequently, 15 μL of derivatization reagent (PITC) should be added to the residue. The mixture was then subjected to a drying process, followed by a 20 min rest period. The result was diluted with 250 μL of dilution reagent (i.e., disodium phosphate buffer, PO_4_HNa_2_, 5 nM, pH 7.4 with 5% CAN) and subjected to centrifugation. Subsequently, 200 μL of the centrifugate was collected, and 20 μL was injected into the HPLC.

The following conditions were employed for HPLC: phase A: 70 mM sodium acetate buffer, pH 6.55, with 2.5% acetonitrile; phase B: 450 mL acetonitrile, 400 mL water, and 150 mL methanol; column: The Movapack C-18 (Waters, S.A.) was a 300 mM × 3.9 i.d. (particle size five μM) apparatus. The precolumn was Novapack 20 mM. The temperature was as follows: 52 °C with a flow of 1 mL min^−1^. The detection was monitored in an UV detector at 254 nM.

### 2.10. Nutritional Quality Indexes of the Fillet

#### 2.10.1. Determination of Fatty Acid in Fillet

To the fatty acid profile, fat was extracted using the adapted protocol of O’Fallon et al. [69], and each fatty acid amount was obtained by gas chromatography. An aliquot of 20 mg from the ground sample was weighed according to the % raw fat and placed in a 15 mL Pyrex tube with a Teflon stopper. Then, 0.7 mL of KOH (potassium hydroxide) 10 N, 5.3 mL of MeOH (methanol), and 1 mL of a solution composed of 0.5 mg mL^−1^ of C13:0 in MeOH were vortexed. The sample was brought to 55 °C for 1.5 h and stirred vigorously for 5 s every 20 min. The sample was then cooled with water at room temperature, and 0.58 mL of H_2_SO_4_ 24 N was added. The sample was brought back at 55 °C for 1.5 h, stirring each vigorously for 20 min for 5 s. Then, it was cooled with water to room temperature, and 1.5 mL of hexane was added, followed by vortexing for 5 min. For 5 min, it was centrifuged through a machine at 3000 revolutions per minute, after which the liquid above the solid material was collected. The measurement of fatty acids was performed using GC software, following the specifications for a Supelco column SP-2560. (Merck KGaA, Darmstadt, Germany).

#### 2.10.2. The Lipid Quality Indexes

The Lipidic Quality Index were calculated using the following equations [72]:

Atherogenicity Index (AI)AI = (12:0 + (4 × 14:0) + 16:0)/(∑MUFA + ∑PUFA n − 6 + ∑PUFA n − 3)

The AI considers the sum of the major saturated and unsaturated fatty acids. In general, saturated fatty acids have some proatherogenic effects, which could favour the adhesion of lipids to the cellular units of the circulatory and immune systems. While unsaturated fatty acids are considered antiatherogenic, reducing the levels of esterified fats, cholesterol and phospholipids, as well as inhibiting plaque aggregation at the arterial level and preventing macro- and microcoronary disease [11,72,73,74,75,76,77].

Thrombogenicity Index (TI)TI = (14:0 + 16:0 + 18:0)/(0.5 × ∑MUFA + 0.5 × (PUFA n − 6 × PUFA n − 3) + (PUFA n − 3/PUFA n − 6))

The TI has been identified as a key factor in the process of clot formation within blood vessels. A relationship has been established between prothrombogenic fatty acids, more commonly referred to as saturated fats, and antithrombogenic acids, which include monounsaturated fats (MUFA, n-6 PUFA y n-3 PUFA) [11,72,73,74,75,76,77].

Cholesterolenic Index (h/H)h/H = (18:1n − 9 + 18:2n − 6 + 20:4n − 6 + 18:3n − 3 + 20:5n − 3 + 22:5n − 3 +22:6 n − 3)/(14:0 + 16:0)

The h/H Index is indicative of the ratio of hypocholesterolemic and hypercholesterolemic fatty acids. The range of indexes in fish is from 1.54 to 4.83. These indexes may be affected by the provision of aquaculture fish diets with higher concentrations of SFA, or by higher consumption of algae in wild species, resulting in values below the range [78,79,80,81,82].

Food Lipid Quality (FLQ)FLQ = 100 × (EPA + DHA)/Total lipids

This index has been employed to determine the degree of correlation between omega-3 polyunsaturated fatty acids (EPA and DHA) and total lipids. The FLQ Index was originally developed as a means to assess the quality of lipids in fish [76,82,83], and meat products [74,84]. Elevated values of the index are indicative of a higher quality of lipids in the diet. However, these values can vary depending on factors such as age, season of harvest, presence of dietary substances, and habitat [74,75,82,84].

Linoleic acid/α-Linolenic Acid Index (LA/ALA)LA/ALA = (C18:2n − 6)/(C18:3n − 3)

The LA/ALA ratio has been demonstrated to indicate a moderate improvement in omega-3 (n − 3) fatty acid levels (EPA, DHA, DPA), with a greater impact on the diet and health of children than adults [82,85].

### 2.11. Fish Purchase Intention

For the purposes of this analysis, which precedes the sensometric study by Calanche et al., [86], a supplementary questionnaire (Table 4) was administered to the judges. This questionnaire collected information regarding which samples the judges would be willing to purchase, their awareness of organic food, their opinion on it, and their willingness to pay a higher price for any of the selected sea bass (organic and conventional).

### 2.12. Economic Analysis

A cost–benefit analysis of the present study was conducted by determining the Economic Conversion Ratio (ECR) and the Economic Profit Index (EPI) adapted from Martínez-Llorens et al. [87] and Jauralde et al. [88]. The total price of the diet per kilogram was calculated, taking into account the value of the referential cost of each ingredient per metric ton (MT).E.C.R = (EUR kg^−1^) = FCR × Diet PriceE. P. I. = Sea bass price − ECR

### 2.13. Statistical Analysis

Growth data, biometric indices, fatty acids of diets were subjected to multivariable analysis of variance (ANOVA), taking the initial weight as a covariate in the study of the final average weight and TCI. The Newman-Keuls test was used to observe significant differences between feeding with a level of significance *p* < 0.05, corresponding to a range of 95% confidence. The statistical programme was used was Statgraphics Plus 5.1 the results were expressed as the mean ± the standard error of the mean.

### 2.14. Ethical Aspects

The experimental protocol was carried out following the European Union Council Directive 2010/63/EU, and Spanish state law (Spanish Royal Decree 53/2013) on the protection of animals used for research scientific after evaluation and approval by the Animal Ethics and Welfare Committee of the Polytechnic University of Valencia (UPV).

Fish in the tanks were checked daily. Also, fish were weighed individually every four weeks, and their health status was assessed through observation, after sedation with clove oil dissolved in water (0.01 mg L^−1^ of water) to minimize animal suffering. Animals were euthanized by an excess of clove oil (150 mg L^−1^) and then dissected.

## 3. Results

### 3.1. Growth and Nutritional Parameters

During the 196-day study period, a significant increase in weight was observed in all diets, although diet 25 ECO originated a lower weight since three first months (Figure 1).

As demonstrated in Table 5, some statistical differences were observed among the ecological diet treatments and the control diet with respect to the FW, SGR, FI, and FCR. Fish fed the control diet (30 CO) exhibited higher final weight and higher specific growth rate (SGR), reaching values of 340 g and 1.09% day^−1^ respectively, but differences only appeared with diet 25 ECO, 304 g and 1.04% day^−1^. Conversely, the lowest FI and FCR indices were observed in fish fed the 30 CO diet, which exhibited a FI of 1.19 g 100 g fish^−1^ day^−1^ and a FCR of 1.47, only different from 25 ECO, 1.35 g 100g fish^−1^ day^−1^ and 1.73, respectively.

### 3.2. Biometric Parameters of Sea Bass Fed Experimental Diets

As illustrated in Table 6, the biometric indices for the final period are presented.

A statistically significant difference was observed for the condition factor (CF), hepatosomatic index (HIS), Viscerosomatic index (VSI), and mesenteric fat (MF) (*p* < 0.05). The indexes with lower values corresponded to the 30 CO diet for CF, VSI, and MF (1.33 g cm^−3^; 11.82%; 5.26%, respectively). However, this was not the case for the HIS, whose value was higher (2.16%).

### 3.3. Body Composition and Nutrient Retention Efficiency

The final results of the body composition of sea bass fed the experimental diets, as well as nutrient retention, are presented in Table 7. Dry matter and fat protein increased in final sea bass from initial sea bass, and ash reduced. No statistically significant differences (*p* > 0.05) were observed for dry matter, crude protein, crude fat, and ash between the ECO and CO treatments. The protein, fat and energy retention efficiencies (PPV; PFV; PEV), showed significant differences (*p*< 0.05) between fish fed the ECO and CO diets, demonstrating that the high values for PPV and PFV indexes were found for the 30 CO diet (25.34% and 76.34%, respectively), and the lowest value was for the 25 ECO diet (21.89% and 62.60%, respectively). However, as for PEV, the highest value was obtained by those fish fed with the 35 ECO diet (30.17%), while the lowest value was represented by the 25 ECO diet (26.27%)

### 3.4. Amino Acid Retention Efficiency

The results of amino acid retention are shown in Table 8. The study revealed no statistically significant differences in most the essential amino acid retention between fish subjected to the CO and ECO treatments at a significance level of *p* > 0.05. However, a statistically significant difference was observed in methionine retention, with the highest value recorded in fish fed the 25 ECO diet (33.18 mg kg body^−1^ weight), and the lowest value observed in fish fed the 30 CO diet (18.20 mg kg body^−1^ weight). Conversely, for the retention efficiency of non-essential amino acids, only three amino acids exhibited significant differences (*p* < 0.05), glutamine, proline, and serine. The highest values of glutamine and proline were observed in fish fed the 35 ECO diet (26.34, 31.75 g kg^−1^ wet weight, respectively), while the lowest values were observed in fish fed the 30 CO treatment (18.76, 21.15 g kg^−1^ wet weight, respectively). In regard to serine retention efficiency, the highest value was observed in fish fed the 25 ECO diet (26.69 g kg^−1^ wet weight), while the lowest value was observed in fish fed the 30 ECO diet (18.82 g kg^−1^ wet weight). No significant differences (*p* > 0.05) were observed for the remaining non-essential amino acids.

### 3.5. Proximal Composition of Fillets, Fatty Acid, and Free Amino Acid Profile

The nutritional composition of the fillets from sea bass fed with ECO and CO feeds is shown in Table 9. No significant differences were observed for total dry matter, protein, and fat, with the exception of ash, which exhibited a higher value in fish fed diets 25 ECO and 30 ECO, while the lowest value was observed in fish fed diet 35 ECO.

A comparative analysis of the fatty acid composition of fillets of sea bass fed different diets (Table 10) revealed that no differences appeared in total saturated, monounsaturated, polysaturated and PUFAs. For monounsaturated fatty acids, samples fed diet 25 ECO demonstrated significantly higher levels (*p* < 0.05) of C17:1, C18:1n-9t, and C20:1. Conversely, diet 30 CO fillets exhibited elevated levels of fatty acids C14:1 and C16:1. In long-chain fatty acids (PUFAs), diet 25 ECO exhibited higher values for LA, ALA, C20:2, and C 20:4n-6, while diet 30 CO demonstrated a higher value for C 22:4n-6. No differences were obtained in EPA, DHA, n-3 HUFAs, n-3/n-6 ratio, and EPA/DHA.

Table 11 presents the free amino acids detected in sea bass fillets from fish fed on the ECO and CO diets. The analysis revealed that the most of free amino acids in fillets were higher in sea bass fed with the 35 ECO, TAU, LYS, GLN, SER, ARG, LEU, PHE, and TYR On the other hand, CYS, MET, TRP, BAL, and ANS levels were higher in the fillets from the 25 ECO treatment group, while HIS, THR and PRO in the sea bass fillets fed the Control 30 CO diet.

### 3.6. Nutritional Quality Indexes of the Fillet

No significant differences were observed for several indexes between the ECO treatments (Table 12).

### 3.7. Purchase Intention

As demonstrated in Table 4, the data was collated from 100 panellists, and a Kruskal–Wallis test with a 95% confidence level was employed to estimate the preference for sea bass fillets fed with the ECO and CO diets. The results of this test are presented in Table 13. A greater preference was expressed for the purchase of sea bass fillets from treatment 30 ECO, while for fillets from treatment 25 ECO, the opposite was true. Conversely, Figure 2 demonstrates that the B-plot design strengthens this inclination in purchase intention, as determined by the Kruskal–Wallis test. Conversely, Figure 3 illustrates the extent of awareness regarding organic food, with a notable inclination towards the following statements: The present study posits that the adoption of environmentally friendly practices, a greater focus on animal welfare, and enhanced availability are imperative for the advancement of the field. In the subsequent question of the survey, 52% of respondents expressed a willingness to incur a higher cost for organic food products. As demonstrated in Figure 4, the willingness to pay is shown to be relatively consistent across the three categories, with no significant differences observed (*p* > 0.05) when classified according to the total number of panellists. The distribution of the categories is as follows: Category 1 accounts for 25%; Category 2 accounts for 41%; and Category 3 accounts for 34%.

### 3.8. Economic Analysis

A cost–benefit analysis of the present study was conducted, and the results are presented in Table 14. With regard to the economic implications of feed, no substantial differences were identified in economic conversion ratio among the several ECO diets (EUR 1.62–1.69 kg^−1^); however, the diet 30 CO, with a cost of EUR 1.29 kg^−1^, was determined to be the most economical option. With regard to the Economic Conversion Ratio (ECR), some differences appeared, being the lowest value for 30 CO (EUR 1.91 kg^−1^) and the highest for 25 ECO (EUR 2.94 kg^−1^). The economic profit index (EPI) exhibited a high value for 30 ECO and 35 ECO diet, reaching EUR 8.41 and 8.42 kg^−1^, while the lowest value was registered for the 30 CO diet, at EUR 3.60 kg^−1^, as consequence of sea bass sell price (EUR 5.50 kg^−1^ for conventional sea bass and EUR 11.00 kg^−1^ for organic sea bass).

## 4. Discussion

### 4.1. Fish Performance

In general, fishmeal and fish oil are considered as complete ingredients for aquaculture feed because they present a good profile of amino acids, polyunsaturated fatty acids and minerals. The challenges when replacing these ingredients is to find that they contain the same properties, and that meet the nutritional needs of the species. It should be noted that fishmeal is very palatable, which is not the case with vegetable sources, because they have low palatability, they are deficient in amino acids, and they have an endless number of anti-nutritional factors depending on the ingredient [54]. In current trial, any reduction on feed ingestion was observed in ECO diets, in the case of 30 ECO growth was similar to the control diets (30 CO) and in 25 ECO the feed intake (FI) was highest. Compared to other research in sea bream, sea bass and trout, it was confirmed that there is a higher feed intake in organic diets compared to conventional diets and that it could be deduced from the absence of synthetic amino acids by trying to compensate with a higher intake [16,37,50]. Nevertheless, the inclusion of high levels of plant ingredients (25 ECO) resulted in a poorer growth and food conversion, and this observation is consistent with the results of other studies that have incorporated organic subproducts of animal origin into the diet, replacing conventional fishmeal. However, these studies have not yielded superior results compared to the control diet with 30% FM [16,35,36].

In sea bass, good results have been obtained when using feed with high degrees of substitution of fishmeal by vegetable protein mixtures in a conventional feed [96,97]. There are few studies where growth is currently evaluated in sea bass fed with organic feed with high substitutions of fishmeal and fish oil. Some previous studies with sea bass compare conventional and organic feed. However. they cannot be compared with these results since the organic feed presented high amounts of fishmeal (56%) compared to conventional (20%), resulting in better growths and feed conversion rates in fish fed with organic feed [50]. Similar studies carried out in other species. the organic feed also presented better growth than conventional feed, without showing significant differences. On the other hand, as did the work of Di Marco et al. [50] in the case of sea bream, the organic feed was also formulated with a higher percentage of fishmeal (63%) than the conventional one (50%) [41]. Another study with organic peameal and organic rapeseed vegetable protein concentrate (PPC) as substitutes for fishmeal in rainbow trout diets, at substitution levels of 0, 16, 31, and 47%, reported no significant differences in the specific growth rate (SGR) and feed conversion rate (FCR). This outcome is attributed to the fact that up to 47% of rapeseed protein concentrate (PPC) while still maintaining a balanced profile of essential amino acids in the feed [98].

As stated in previous works, the utilization of organic animal by-products as substitutes for conventional fishmeal has been demonstrated to have a substantial impact on the circular economy and to promote the incorporation of organic ingredients. However, these by-products have been observed to exert an adverse effect on growth and final weight [16], which in this study demonstrated minimal disparities in weight gain between ECO and CO diets at an inclusion level of 30% of FM. This ensured that parameters such as feed intake and Feed Conversion Ratio (FI, FCR) would be similar.

### 4.2. Final Biometric Parameters

Sea bass fed control diet 30 CO presented a lower Viscerosomatic index (11.8%) than ECO diets (13.33–13.83%) which agrees with others studies with a high substitution of fishmeal by ecological vegetable sources resulted in a significant increase in the VSI that ranged between 8.39 and 14.0%, and the same happened with the HSI high values 1.12 and 2.64% [50,99].

A study by Gaylord et al. [100] identified a correlation between dietary methionine deficiency and intraperitoneal fat increment, which present study has demonstrated, because all organic diets exhibit a deficiency of this essential amino acid, and an increment of visceral or mesenteric fat (MF) and Viscerosomatic index (VSI) (Table 5). However, the incorporation of methionine of vegetable origin in studies involving sea bream and sea bass, with diets comprising trout and sea bass by-products as hydrobiological ingredients, without the presence of conventional FM, has emerged as a promising alternative. This approach holds significant potential for the revaluation of fishery by-products and the promotion of sustainability. Consequently, it necessitates a re-examination of the optimal proportions of alternative protein ingredients to be incorporated into the formulation, with the objective of meeting the nutritional requirements of the European sea bass [36,101].

### 4.3. Body Composition and Retention Nutrient Efficiency

In regard to body composition, no statistically significant differences (*p* > 0.05) between the ECO and CO group diets were observed with respect to dry matter, crude protein, crude fat, and ash, corroborating the findings reported in European sea bass by Tefal et al. [35], who utilized vegetable ingredients and organic animal by-products (soybean meal, Iberian pork, insect meal, trout by-product) without fishmeal. Conversely, other studies on organic diets for sea bream exhibited considerably lower outcomes concerning body composition levels, this can be attributed to the incorporation of poultry meal, which is known to reduce body fat content [36,102]. However, Sabbagh et al. [103] achieved favourable outcomes without compromising growth and protein quality. In the present study, the protein and fat retention efficiency (PPV, PFV) exhibited superiority over the findings reported by Tefal et al. [35].

Nutrient retention efficiency showed variability among treatments, with fish fed organic diets having lower protein productive value (PPV) rates. This phenomenon can be attributed to a deficiency of essential amino acids (EAA), which limited the synthesis of proteins in the fish. In addition, Fat Productive Value (FPV) and Energy Productive Value (EPV) were significantly increased in fish fed ECO diets, especially in the 35 ECO treatment. This increase reflects a significant crude fat and energy content in the final body composition of the fish due to alterations in lipid metabolism. Such alterations are associated with inadequate lipid distribution at the tissue level, thereby compromising the activity of lipogenic enzymes [16,104,105].

The former utilized organic subproducts of animal origin in sea bass feed, yielding ranges of 17 to 12% for PPV and 65 to 56% for PFV. In contrast, the present study employed a greater quantity of organic ingredients of vegetable origin (spelt bran, corn meal, and soybean meal), yielding ranges of 25 to 22% for PPV and 76 to 62% for PFV. This discrepancy can be attributed to the distinct composition of the organic ingredients utilized in each study. On the other hand, the use of these alternative plant ingredients is promising at a 30% fishmeal inclusion level, as reflected in the present study. However, high substitutions by other organic animal sources in diets for carnivorous fish could negatively affect growth, protein productivity, and retention values of essential amino acids [36].

### 4.4. Retention Amino Acid Efficiency

Only a significant difference was observed in the efficiency of retention of methionine, the 25 ECO diet, which incorporated 25% fishmeal (FM), demonstrated the highest methionine retention rate of 33.18%, indicating a notable increment in retention efficiency in relation the others ECO diets (25.9%) and control diet 30 CO (18.2%). These differences can be attributed to the need, for compensating the dietary methionine deficiency, (0.7–0.9% in ECO diets, and 1.25% in control diet). This compensation was effective in the case of sea bass fed 20 ECO and 35 ECO, which presented a similar growth than control, but sea bass fed 25 ECO was no able to reach the same weight. Likewise, as indicated Tefal et al. [16], European bass fed organic diets containing by-products from trout, Iberian pork, and insect meal exhibited significant levels of methionine; despite diets were supplemented, Iberian pork and mix diets presented lowest level of methionine (0.5% compared 0.7–1.0%), and although its retention was higher (24% compared 12–15%) the growth could not be compensated, which agree with diet 25 ECO in current trial. However, the efficiency of the retention of other essential amino acids remained largely unaltered, a property that can be ascribed to the level of incorporation of conventional fishmeal at levels of 25%, 30%, and 35%.

### 4.5. Proximal Composition of Fillets, Fatty Acid, and Free Amino Acid Profile

In the proximal composition (Table 9) of the sea bass fillets fed with ECO and CO feed, no significant differences were found for crude protein, dry matter, and ether extract. However, a lower ash content was observed in sea bass fillets fed with the 35 ECO diet. Conversely, to interpret the proximate composition of the fillets, it is imperative to consider samples of the lipid profile and free amino acids, which can contribute to sensory and quality studies [106].

The fatty acid composition of sea bass muscle is associated with the fatty acid profile of the diets provided. The composition of the fish’s diet can influence its growth and health. The predominant saturated fatty acids are typically C16:0 and C18:0, while monounsaturated fatty acids are principally represented by C18:1n9 [107,108]. A higher presence of these fatty acids has been shown to promote better membrane fluidity when the ambient temperature changes [109]. Conversely, marine fish species lack the capacity to synthesize unsaturated fatty acids (HUFAs) from PUFAs, a deficiency that accounts for the substantial accumulation of linoleic acid (LA) in the fillets of sea bass fed conventional and organic diets. This phenomenon can be attributed to the absence of one or more physiological synthesis processes [110]. Furthermore, the deposition of fatty acids in the musculature of sea bass, for ∑ SFA, ∑ MUFA, and ∑ PUFA, can be influenced by the diet and the nature of the protein and lipid sources, whether conventional or organic (including by-products from agriculture, terrestrial livestock, or aquatic sources). It is imperative to take into account the prior treatment of the specimens, as well as whether they are wild or farmed, given that this can lead to ineffective or efficient distribution of fatty acids in the dorsal or ventral regions of carnivorous fish species such as sea bass (*Dicentrarchus labrax*) and sea bream (*Sparus aurata*) [16,36,76,111,112,113].

In contrast, the variation in the concentration of fatty acids in carnivorous fish, even between species of the same phylum, is influenced by a variety of factors. The primary factor contributing to this imbalance is the absence of the enzyme’s delta-12 desaturase and delta-15 desaturase, which are essential for the synthesis of ALA and LA from C 18:1n-9. The degree of their composition is determined by their ability to elongate and desaturate, thereby becoming long-chain PUFAs [114,115,116]. However, as illustrated in Table 10, a higher concentration of n-6 is observed in fish fed organic diets. This finding suggests that an increased n-6 intake may influence the ratio of PUFA to n-3 fatty acids. As indicated in the works of Monroig et al. [115], Saini et al. [117], Tefal et al. [36], and Glencross et al. [116].

Tefal et al. [36] indicate that when sea bream were fed organic diets that include trout by-products (TRO), sea bass (SBS), and poultry (POU), as well as a mixture of these (MIX) and an ECO diet containing 30% fishmeal, better growth performance is observed in the CO and ECO diets. However, the CO diet exhibited enhanced integration of fatty acids in the fillet, a finding that contrasts with the results at present study of elevated levels of fatty acids in the 35 ECO diets comprising 35% fishmeal. The fish that demonstrated optimal weight gain were those fed the 30 CO and 30 ECO diets, both containing 30% fishmeal. The nature of the ingredients has been demonstrated to influence palatability, resulting in increased intake to meet nutritional requirements. This, in turn, affects the lipid metabolism of fatty acids in fillets on the ECO and CO diets, as indicated by several authors in their studies on fatty acid composition in marine fish [116,118,119,120].

As demonstrated in Table 11, the analysis of marine and continental fish species reveals the presence of essential amino acids, which are essential for a nutritionally balanced diet. However, external factors such as diet, time, processing, and storage at varying temperatures have been demonstrated to exert a substantial influence on the composition of muscle and free amino acids [106,121,122,123,124]. In the case of SER, values were found in sea bass fillets fed with the 35 ECO diet, which determine the sweetness indicator [106,124,125]. Concurrently, in diverse aquatic organisms, certain free amino acids—including ORN, GLU, ASP, ALA, SER, and GLY—play a pivotal a key role in the organoleptic characteristics of food, affecting umami flavour, sweetness, aroma, and bitter-sweet taste [106,124,126]. In addition, some of these compounds are precursors of secondary metabolites that produce volatile compounds [124]. These free amino acids were studied in sea bream [106], herbivorous carp [126] and yellowfin sea bream [127], Barramundi [128], and shrimp [124,129]. In contrast, the present study demonstrated that the ECO and CO treatments exhibited elevated levels of six out of the twenty-six free amino acids, with TAU registering at 370.6 mg 100 g^−1^ (35 ECO), GLY at 56.5 mg 100 g^−1^ (30 ECO), LYS at 67.3 mg 100 g^−1^ (35 ECO), HIS at 51.6 mg 100 g^−1^ (30 CO), ALA at 29.01 mg 100 g^−1^ (35 ECO), and GLU at 28.50 mg 100 g^−1^ (30 ECO).

However, other studies carried out on fillets of gilthead sea bream (*Sparus aurata*) (whole fish, gutted, and filleted) that were frozen reported levels of TAU ≥ 150 mg 100 g^−1^ [106]. A paucity of research was identified on free amino acids in fish fed organic diets, underscoring the significance of this study in treatments applied to sea bass. Consequently, a preceding sensory study [86], published prior to the present study, demonstrated that sea bass fillets fed the 30 ECO diet exhibited optimal sensory attributes, whereas the fillets from the 25 ECO treatment exhibited contrasting outcomes. This divergence may be attributed to the fact that the concentration of each amino acid can affect the sensory quality of sea bass.

### 4.6. Nutritional Quality Indexes of the Fillet

In relation to the nutritional and qualitative indices of the fillets of sea bass that were fed the ECO and CO diets, it was observed that both the nutritional parameters (IA, TI, HH, FLQ) and the qualitative parameters (ω6/ω3, PUFA/SFA, MUFA/SFA, PUFA/MUFA, (PUFA + MUFA)/SFA, and LA/ALA) were found to be within the ranges defined by the references of various authors. Concurrently, research on sea bass indicates that the atherogenic index (AI) values exceed 0.50, while the thrombogenic index (TI) values reach 0.33, across both farmed and wild specimens [130]. Conversely, Monteiro et al. [90] have indicated that the AI ranges in sea bass vary between 0.40 and 0.42, suggesting that this fish exhibits a higher degree of adaptability to the artificial diets provided. Consequently, the quality of the final product may be contingent upon the quantity of vegetable oils incorporated during the manufacturing process, as well as the fatty acid profile present in the raw materials, particularly in the case of organic ingredients. This may result in a decrease in IA, TI, and FLQ levels, evidencing the lack of other parameters that could provide a better understanding of the nutritional value of the fillets and a higher proportion of beneficial fatty acids [82,111,130,131,132]. In contrast, Monteiro et al. [90] reported TI values of 0.191–0.63, which are considerably higher than those observed in the present study (0.00024–0.00038), without exceeding the defined range [11,72,76,89]. It is evident that these values may fluctuate as sea bass specimens attain larger sizes [133].

In the case of h/H, it was observed that the lowest value was recorded for sea bass fillets that were fed the control diet 30 CO (3.40). Conversely, in the ECO diets, the values were elevated for both the fillets from group 25 ECO and group 30 ECO (4.55 and 4.50, respectively). However, Santos-Silva et al., [91] have indicated that index levels may fluctuate without exerting an effect on specific fatty acids in the cholesterol metabolism. This phenomenon can be attributed, in large part, to the type of diet, the feeding time, and the nature of the oils used. Consequently, such disparities are discernible in other carnivorous species inhabiting analogous environments. In the case of *Labeo rohita*, when attempting to replace fish oil with peanut oil, up to 60% could be substituted without significantly impairing the growth and nutritional quality of the fillet (FLQ) [134], *Merluccius gayi* a value of 2.23 is reported, for *Seriola lalandi* a value of 2.14 [81], while for *Salmon trutta* a range of 1.88 to 2.16 is established [78].

In the present study, no significant differences were observed in FLQ between the ECO and CO diets (8.97–8.63), indicating minimal values, attributable to the organic sources and the incorporation of 50% fish oil and soybean oil. The primary objective of this index is to assess the nutritional quality of the fillet, a particularly salient aspect in the context of marine species. It has been demonstrated that an elevated proportion of fish oil in the diet is associated with enhanced nutritional quality and elevated FLQ values, attributable to an increase in the incorporation of EPA and DHA [82,84]. For *Labeo rohita*, a superior nutritional quality index (32.70) was observed in diets containing 100% fish oil. A decrease in fish oil and the introduction of an alternative energy source have been demonstrated to result in a reduction in FLQ values [134]. In the case of trout, a value of 17.97 was reported [74], and for sea bream, ranges of 23.7 to 13.5, these values being lower than those of feeds that did not contain fish oil [84,135].

However, the other indices remain within the established range [92,93]. A higher ratio of omega-6 to omega-3 fatty acids indicates a higher incorporation of vegetable oils or oilseed meal, which may cause an imbalance in the sources of polyunsaturated fatty acids (PUFAs), as well as in the concentration of eicosapentaenoic acid (EPA) and docosahexaenoic acid (DHA) in the fillets. A notable increase in the omega-6/omega-3 ratio has been observed in sea bream and sea bass, with values rising from 0.22 to 0.26 and from 0.44 to 0.48, respectively [130]. In a separate study that utilized organic diets, it was observed that the productive value of fatty acids in fillets of sea bream fed with certified organic animal by-products (trout, sea bass, poultry, MIX) [36] was also affected, both in terms of growth and the deposition of lipids in the fillets. In contrast, it has been posited that the ratio of PUFA/SFA in the human diet should exceed 0.45 [93], a figure that exceeds the values found in this study on sea bass fillets that received ECO and CO diets (1.60 to 2.06) [82,92].

### 4.7. Purchase Intention

The sea bass fed a diet containing 30% fishmeal (30ECO) was the most preferred, while the sea bass fed a diet containing 25% fishmeal (25ECO) was the least preferred. This was due to changes in their organoleptic characteristics, small but substantial variations in the free amino acids GLY and GLU, which were responsible for the seafood and umami flavours that influenced this choice. Meanwhile, according to the panellists, the concept of organic food refers more to food that is “environmentally friendly,” “animal welfare-friendly,” and “more readily available,” thereby maintaining a somewhat clear vision of sustainable production. The majority of panellist (Figure 4) indicated their willingness to pay between EUR 0.80 and 1.76 more for organic sea bass than for conventional sea bass (EUR 5.50 per kg). The amount with the highest percentage of panellists willing to pay the reference price was EUR 1.51 (41%), resulting in a tentative sale price for organic sea bass fillets of EUR 7.0 per kg.

### 4.8. Economic Analysis

The price of organic diets and the economic conversion ratio were higher in ECO diets than conventional diet, due the cost of organic ingredients, which could be an inconvenient for organic production. Nevertheless, the economic profit index also was higher with organic diets, due the higher sell price of organic sea bass, EUR 11 per kg respect EUR 5.5 per kg, but the broken price would be EUR 7 per kg, price more affordable for consumers on the other hand, the organic label would open new fish market, interested in this ecological production.

## 5. Conclusions

The present study demonstrates the feasibility of feeding European sea bass with organic diets containing vegetable ingredients with varying levels of fishmeal. Optimal results in terms of growth performance, FCR, PPV, PFV, PEV, fillet quality, and economic profit were achieved with the diet containing 30% fishmeal and organic plant ingredients (30 ECO), without compromising the nutritional quality indices of the fillet. However, it should be noted that these outcomes may vary depending on the origin and composition of the organic ingredients used. Nevertheless, this study offers an optimistic perspective for future research on organic aquaculture, allowing for the optimization of feed and enhanced utilization of natural resources and the production of high-quality, sustainable aquatic food products.

## Figures and Tables

**Figure 1 animals-15-02339-f001:**
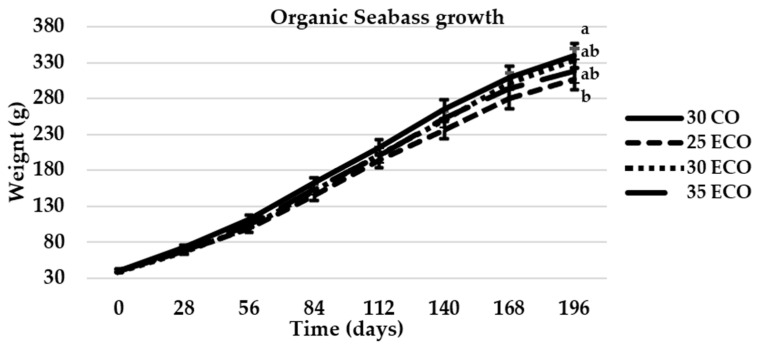
Mean live weight of sea bass evolution during the 196-day experiment. The values that were represented as means (n = 3) exhibited different letters in each sampling, indicating significant differences (*p* < 0.05).

**Figure 2 animals-15-02339-f002:**
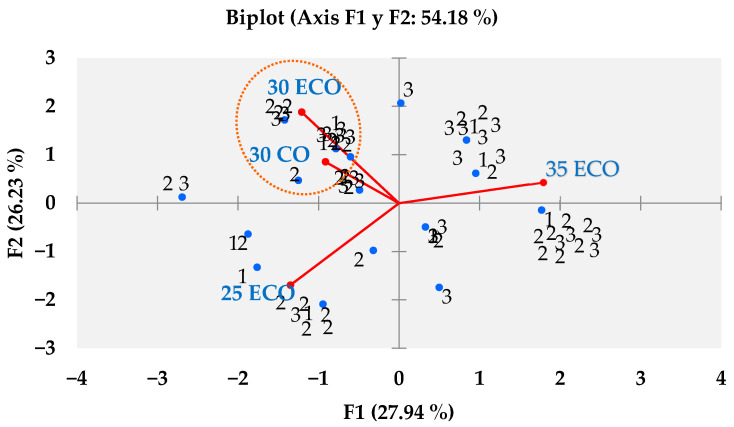
B-plot design of preference in purchase intention for sea bass fillets fed diets 25 ECO, 30 ECO, 35 ECO, and 30 CO.

**Figure 3 animals-15-02339-f003:**
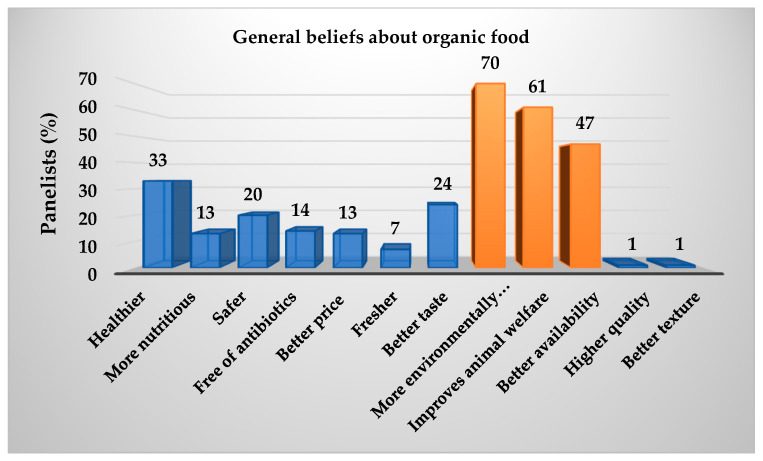
Panellists’ overall knowledge of organic food.

**Figure 4 animals-15-02339-f004:**
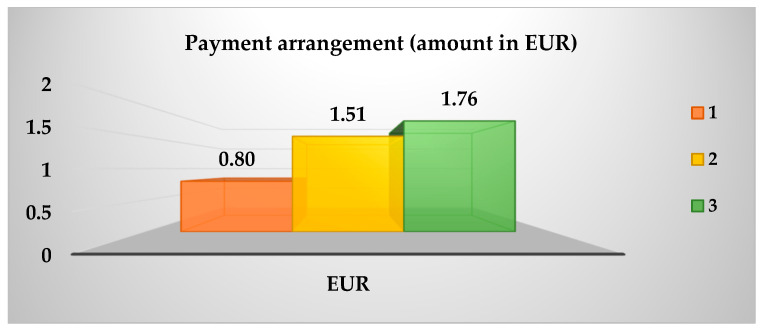
Additional amount (EUR) willing to pay in the price.

**Table 1 animals-15-02339-t001:** Formulation and nutritional composition of experimental feed.

Diets:	25 ECO	30 ECO	35 ECO	30 CO
Ingredients (g kg^−1^)				
Fishmeal	250	300	350	300
Wheat	-	22	47	179
Wheat gluten	-	-	-	122
Spelt bran	-	10	20	-
Corn	-	8	15	-
Soya meal	586	504	420	218
Soybean oil	55	59	62	72
Fish oil	72	67	63	71
Calcium Phosphate	27	20	13	20
Taurine	-	-	-	5
Methionine	-	-	-	3
Vitamins ^1^	10	10	10	10
Nutritional composition (% dry matter)				
Dry matter	92	92	92	92
Crude protein	46.1	46.7	47.1	47.2
Crude fat	17.1	17.1	16.8	15.4
Ash	9.3	9.2	9.2	8.1
Gross energy (MJ)	24.2	24.0	24.1	24.3

^1^ Mix of vitamin and mineral supplementation (g kg^−1^): premix, 25; Choline, 10; DL-a-tocopherol, 5; ascorbic acid, 5; (PO_4_)_2_Ca_3_, 5. Premix composition: retinol acetate, 1,000,000 IU kg^−1^; calciferol, 500 IU kg^−1^; DL-a-tocopherol, 10; sodium bisulfite menadione, 0.8; thiamine hydrochloride, 2.3; riboflavin, 2.3; pyridoxine hydrochloride, 15; cyanocobalamin, 25; nicotinamide, 15; pantothenic acid, 6; folic acid, 0.65; biotin, 0.07; ascorbic acid, 75; inositol, 15; betaine, 100; polypeptides 12.

**Table 2 animals-15-02339-t002:** Amino acid composition of experimental diets.

	DIETS			
AA	25 ECO	30 ECO	35 ECO	30 CO
Essential amino acids (g 100 g^−1^ wet weight)				
Arginine	2.74	2.57	2.7	2.58
Histidine	0.90	0.94	1.00	0.74
Isoleucine	1.72	1.71	1.69	1.58
Leucine	2.96	2.94	2.93	2.82
Lysine	2.54	2.41	2.55	2.00
Methionine	0.70	0.77	0.88	1.25
Phenylalanine	1.77	1.79	1.82	1.76
Threonine	1.54	1.44	1.44	1.34
Valine	1.97	1.97	1.99	1.83
Non-essential amino acids (g 100 g^−1^ wet weight)				
Alanine	1.85	1.90	1.90	1.62
Aspartic acid	4.04	3.69	3.70	2.87
Cysteine	0.41	0.38	0.51	0.36
Glutamic acid	6.22	6.13	6.06	7.86
Glycine	1.90	1.91	2.10	1.79
Proline	1.52	1.50	1.62	2.11
Serine	1.59	2.26	1.80	1.71
Tyrosine	1.19	1.30	1.29	1.18
EAAs/NEAAs	0.90	0.87	0.90	0.83

EAAs, essential amino acids; NEAAs, no essential amino acids.

**Table 3 animals-15-02339-t003:** Fatty acid composition (g 100g^−1^ of diets) of experimental diets.

	DIETS			
FA	25 ECO	30 ECO	35 ECO	30 CO
C 12:0	0.004	0.006	0.004	0.004
C 13:0	0.349	0.349	0.344	0.325
C 14:0	0.244	0.245	0.249	0.239
C 15:0	0.046	0.043	0.044	0.041
C 16:0	2.457	2.372	2.364	2.019
C 17:0	0.068	0.066	0.067	0.063
C 18:0	0.823	0.786	0.763	0.619
C 20:0	0.052	0.050	0.049	0.0385
C 22:0	0.056	0.050	0.049	0.035
C 24:0	0.031	0.032	0.032	0.023
∑ saturated	4.131	4.001	3.967	3.408
C 14:1	0.004	0.002	0.002	0.002
C 16:1	0.281	0.280	0.286	0.277
C 17:1	0.035	0.030	0.035	0.035
C 18:1n-7	0.542	0.529	0.537	0.463
C 18:1n-9t	0.018	0.017	0.017	0.016
C 18:1n-9c	3.231	3.159	3.029	2.652
C 20:1	0.187	0.179	0.178	0.188
C 22:1n-9	0.038	0.034	0.037	0.036
C 24:1	0.063	0.057	0.068	0.063
∑ monounsaturated	4.398	4.285	4.188	3.732
C 18:2n-6c	5.425	5.359	5.034	4.212
C 18:3n-6	0.017	0.015	0.016	0.025
C 18:3n-3	0.949	0.932	0.862	0.646
C 20:2	0.114	0.112	0.118	0.112
C 20:3n-6	0.009	0.008	0.009	0.009
C 20:3n-3	0.023	0.022	0.022	0.020
C 20:4n-6	0.087	0.085	0.087	0.086
C 22:2	0.042	0.043	0.043	0.042
C 20:5n-3 EPA	0.382	0.397	0.421	0.403
C 22:4n-6	0.063	0.060	0.070	0.064
C 22:5n-3	0.117	0.119	0.120	0.119
C 22:6n-3 DHA	1.280	1.355	1.394	1.344
∑ polyunsaturated	8.508	8.507	8.195	7.081
∑ n-6	5.601	5.527	5.216	4.395
∑ n-3	2.752	2.826	2.819	2.533
n-3 HUFAS	1.779	1.871	1.936	1.867
n-3/n-6	0.491	0.511	0.541	0.576
EPA/DHA	0.299	0.293	0.302	0.300

Note: HUFAS, highly unsaturated fatty acids; EPA, eicosapentaenoic acid; DHA, docosahexaenoic acid.

**Table 4 animals-15-02339-t004:** Intent to purchase.

1. Indicate which of the samples you tasted you would be willing to buy			
2. Do you know what an organic food is?			
Yes ( ) No ( )			
3. If you answered yes to the previous question, do you believe that organic food is: (check all that apply):			
Healthier	( )	Higher quality	( )
More nutritious	( )	Better texture	( )
Safer	( )	More environmentally friendly	( )
Fresher	( )	Improves animal welfare	( )
Better taste	( )	Better availability	( )
Better price	( )	Free of antibiotics	( )
4. If you knew that the samples were organic sea bass, would you be willing to pay more?			
yes	( )	No	( )
How much more? _______ euros/kg			

**Table 5 animals-15-02339-t005:** Final results of growth and nutritional use of sea bass fed with the different experimental feeds for 196 days.

	25 ECO	30 ECO	35 ECO	30CO	SEM
Initial weight (g)	40.4	38.6	40.4	40.4	0.5
Final weight (g)	306.5 ^b^	332.8 ^ab^	318.1 ^ab^	340.4 ^a^	7.4
Mortality (%)	3.3	3.6	3.3	3.3	1.16
SGR (% day^−1^) ^1^	1.04 ^b^	1.08 ^a^	1.05 ^ab^	1.09 ^a^	0.01
FI (g 100g^−1^ fish day^−1^) ^2^	1.35 ^b^	1.27 ^a^	1.25 ^a^	1.19 ^a^	0.02
FCR^3^	1.73 ^b^	1.59 ^a^	1.58 ^a^	1.47 ^a^	0.03

The values presented in the table are the mean ± SEM, Standard Error of the Mean (n = 3). Superscripts with different letters indicate significant statistical differences with a *p* < 0.05 Newman-Keuls test. ^1^ SGR = 100 × ln (final weight/initial weight)/days. ^2^ FI (g 100g^−1^ fish day^−1^) = 100 × feed consumption (g)/average biomass (g) × days. ^3^ FCR = feed consumption (g)/biomass gain (g).

**Table 6 animals-15-02339-t006:** Final biometric parameters of sea bass fed with the different experimental feeds for 196 days.

	25 ECO	30 ECO	35 ECO	30 CO	SEM
CF (g cm^−3^) ^1^	1.54 ^c^	1.44 ^b^	1.59 ^c^	1.33 ^a^	0.03
VSI (%) ^2^	13.83 ^b^	13.33 ^b^	13.80 ^b^	11.82 ^a^	0.44
HSI (%) ^3^	1.60 ^b^	1.56 ^b^	1.81 ^b^	2.16 ^a^	0.09
MF (%) ^4^	6.86 ^b^	6.27 ^ab^	7.62 ^b^	5.26 ^a^	0.45

Means of triplicate groups. Values are presented as mean ± SEM (standard error of the pooled means). Values in the same row with different superscript letters are significantly different (*p* < 0.05). ^1^ Condition factor CF = 100 × (total fish weight (g)/total length^3^ (cm)). ^2^ Viscerosomatic index (%) VSI = 100 × (visceral weight (g)/fish weight (g)). ^3^ Hepatosomatic index (%) HSI = 100 × (liver weight (g)/fish weight (g)). ^4^ Mesenteric fat (%) MF = 100 × (mesenteric fat weight (g)/fish weight (g)).

**Table 7 animals-15-02339-t007:** Whole-body composition and retention efficiency of sea bass at the end of the experiment.

	INITIAL	25 ECO	30ECO	35 ECO	30 CO	SEM
MS %	26.12	37.36	36.37	37.97	36.91	0.50
PB%	16.12	17.29	17.23	16.99	17.45	0.24
GB%	4.66	16.65	15.59	17.74	15.86	0.58
CE %	5.48	3.50	3.62	3.58	3.72	0.32
Energy (kj g^−1^)	19.47	27.38	26.83	28.10	26.73	0.39
PPV (%) ^1^		21.89	23.42 ^ab^	22.97 ^b^	25.34 ^a^	0.70
PFV (%) ^2^		62.60 ^b^	63.17 ^b^	74.23 ^a^	76.34 ^a^	3.24
PEV (%) ^3^		26.27 ^b^	27.23 ^ab^	30.17 ^a^	29.24 ^ab^	1.06

The values presented in the table are the mean ± SEM. (n = 3). Superscripts with different letters indicate differences statistically significant with a *p* < 0.05 by the Newman-Keuls test. ^1^ PPV (%) = Retained protein (final fish protein × Final biomass (g)) × 100 − Initial fish protein × initial biomass (g)/Protein ingested (Kg of food ingested × % crude protein of feed). ^2^ PFV (%) = Retained fat (final fish fat × Final biomass (g)) × 100 − Initial fish fat × initial biomass (g)/Fat ingested (Kg food ingested × % crude fat of feed). ^3^ PEV (%) = Energy retained (Final fish energy × Final biomass (g)) × 100 − Initial fish energy × initial biomass (g)/energy ingested (Kg feed ingested × % feed energy).

**Table 8 animals-15-02339-t008:** Sea bass amino acid retention efficiency (AARE) at the end of the experiment.

	25 ECO	35 ECO	30 ECO	30 CO	SEM
EAA (g kg^−1^ wet weight)					
Arginine	30.24	34.33	32.14	31.40	2.25
Histidine	20.30	18.28	20.03	23.16	2.15
Isoleucine	26.79	27.34	26.57	27.93	2.42
Leucine	21.86	23.19	19.98	22.31	1.82
Lysine	28.82	30.13	30.31	32.78	3.80
Methionine	33.18 ^a^	25.92 ^b^	25.92 ^b^	18.20 ^c^	1.10
Phenylalanine	19.66	20.12	18.33	18.38	1.80
Threonine	24.73	28.20	27.20	28.66	2.01
Valine	24.63	25.06	22.10	25.02	1.83
NEAA (g kg^−1^ wet weight)					
Alanine	32.37	34.62	29.04	35.67	1.78
Aspartate	25.57	27.68	25.75	31.57	2.53
Cystine	16.42	13.31	15.42	20.37	2.21
Glutamine	25.31 ^a^	26.34 ^a^	22.13 ^ab^	18.76 ^b^	1.63
Glycine	44.53	48.22	40.51	43.44	3.12
Proline	29.16 ^ab^	31.75 ^a^	30.20 ^ab^	21.15 ^b^	2.78
Serine	26.69 ^a^	23.99 ^a^	18.82 ^b^	22.89 ^ab^	1.34
Tyrosine	22.10	20.56	22.48	20.98	2.68

Data are expressed as means ± SEM (n = 3). Means with different superscript letter in the same row significantly differ at *p* < 0.05. EAA: Essential Amino Acids. NEAA: Not Essential Amino Acids.

**Table 9 animals-15-02339-t009:** Proximal composition of fillets from sea bass fed the experimental diets.

DIETS	25 ECO	30 ECO	35 ECO	30 CO	SEM
Dry matter (%)	28.2	28.11	30.68	28.88	0.93
Ash (%)	6.35 ^a^	6.07 ^a^	4.53 ^b^	5.68 ^a^	0.36
Crude protein (%)	67.35	71.4	65.36	69.99	2.44
Ether extract (%)	27.75	24.13	32.8	27.17	2.53

Data in the same row with different superscripts differ at *p* < 0.05. (n = 9).

**Table 10 animals-15-02339-t010:** Fatty acid profile (g 100g^−1^) of fillets from sea bass fed organic diets.

	DIETS				
AG	25 ECO	30 ECO	35 ECO	30 CO	SEM
C 12:0	0.005	0.004	0.005	0.005	0.001
C 13:0	0.804 ^ab^	0.648 ^b^	0.877 ^a^	0.733 ^ab^	0.062
C 14:0	0.451	0.373	0.469	0.444	0.035
C 16:0	4.464 ^ab^	3.642 ^b^	4.753 ^a^	4.608 ^ab^	0.377
C 18:0	1.357	1.124	1.395	1.349	0.106
C 20:0	0.061 ^ab^	0.050 ^b^	0.066 ^a^	0.060 ^ab^	0.004
C 22:0	0.047 ^a^	0.039 ^ab^	0.046 ^a^	0.037 ^b^	0.003
C 24:0	0.027 ^b^	0.025 ^b^	0.034 ^a^	0.030 ^ab^	0.002
∑ saturated	7.401	6.051	7.831	7.433	0.602
C 14:1	0.005 ^ab^	0.004 ^b^	0.005 ^ab^	0.006 ^a^	0.001
C 16:1	0.665 ^ab^	0.514 ^b^	0.658 ^ab^	0.690 ^a^	0.057
C 17:1	0.084 ^a^	0.065 ^b^	0.079 ^ab^	0.077 ^ab^	0.006
C 18:1n-7	1.120	0.920	1.089	0.983	0.084
C 18:1n-9c	6.946 ^ab^	5.547 ^b^	7.199 ^a^	6.328 ^ab^	0.552
C 18:1n-9t	0.062 ^a^	0.038 ^b^	0.058 ^a^	0.041 ^b^	0.004
C 20:1	0.582 ^a^	0.411 ^b^	0.498 ^ab^	0.450 ^ab^	0.053
C 24:1	0.116 ^a^	0.097 ^b^	0.117 ^a^	1.101 ^ab^	0.006
∑ monounsaturated	9.673	7.674	9.788	9.749	0.768
C 18:2n-6c LA	9.153 ^a^	7.434 ^ab^	8.909 ^a^	6.733 ^b^	0.620
C 18:3n-3 ALA	1.406 ^a^	1.125 ^ab^	1.302 ^a^	0.914 ^b^	0.107
C 18:3n-6	0.082	0.065	0.076	0.08022	0.007
C 20:2	0.422 ^a^	0.328 ^b^	0.371 ^ab^	0.319 ^b^	0.031
C 20:3n-3	0.044	0.037	0.040	0.035	0.003
C 20:3n-6	0.019	0.018	0.020	0.018	0.002
C 20:4n-6 AA	0.187 ^a^	0.158 ^b^	0.189 ^a^	0.178 ^ab^	0.009
C 20:5n-3 EPA	0.675	0.547	0.694	0.632	0.006
C 22:2	0.091	0.073	0.088	0.080	0.007
C 22:4n-6	0.106 ^ab^	0.089 ^b^	0.115 ^ab^	0.117 ^a^	0.009
C 22:5n-3	0.243	0.189	0.236	0.213	0.019
C 22:6n-3 DHA	2.786	2.270	2.863	2.612	0.211
∑ polyunsaturated	15.214	12.330	14.903	11.929	1.031
∑ n-6	9.546	7.763	9.309	7.125	0.647
∑ n-3	5.154	4.167	5.135	4.405	0.346
n-3 HUFAS	3.704	3.006	3.793	3.457	0.235
n-3/n-6	0.540	0.537	0.552	0.618	0.535
EPA/DHA	0.242	0.241	0.242	0.242	0.026

HUFAS, highly unsaturated fatty acids; EPA, eicosapentaenoic acid; DHA, docosahexaenoic acid; AA, arachidonic acid; ALA, alpha-linolenic acid; LA, linoleic acid. Means with different superscript letter in the same row significantly differ at *p* < 0.05.

**Table 11 animals-15-02339-t011:** Free amino acid profile (mg 100g^−1^) in fillets from sea bass fed with the experimental diets ECO and CO.

		DIETS				
Free AA		25 ECO	30 ECO	35 ECO	30 CO	SEM
TAU	Taurine	338.438 ^ab^	312.774 ^b^	370.601 ^a^	365.088 ^ab^	17.635
GLY	Glycine	50.661	56.558	52.336	55.888	4.568
LYS	Lysine	62.734 ^ab^	42.609 ^bc^	67.352 ^a^	22.186 ^c^	7.117
HIS	Histidine	20.404 ^c^	37.763 ^ab^	24.746 ^bc^	51.614 ^a^	4.674
ALA	Alanine	25.704	27.038	29.008	24.753	1.739
GLU	Glutamic acid	23.199	28.501	24.560	26.771	1.812
CYS	Cysteine	14.869 ^a^	10.496 ^b^	10.447 ^b^	11.893 ^b^	0.677
GLN	Glutamine	9.141 ^b^	8.559 ^b^	19.4 ^a^	6.841 ^b^	2.331
ORN	Ornithine	7.207	8.251	7.813	6.561	1.301
SER	Serine	7.62 ^a^	7.505 ^a^	8.299 ^a^	5.806 ^b^	0.565
THR	Threonine	5.39 ^bc^	7.06 ^ab^	4.639 ^c^	7.763 ^a^	0.657
ARG	Arginine	6.079 ^b^	6.579 ^ab^	9.16 ^a^	3.141 ^c^	0.914
LEU	Leucine	4.007 ^ab^	3.089 ^bc^	4.387 ^a^	2.692 ^c^	0.428
HXP	Hydroxyproline	2.512 ^b^	4.594 ^a^	3.772 ^ab^	4.34 ^a^	0.528
PRO	Proline	2.829 ^b^	2.579 ^b^	2.334 ^b^	6.663 ^a^	0.694
VAL	Valine	2.769 ^ab^	2.505 ^b^	3.298 ^a^	2.282 ^b^	0.248
ILE	Isoleucine	2.081	2.529	2.564	3.469	0.636
MET	Methionine	2.313 ^a^	2.03 ^ab^	2.291 ^ab^	1.818 ^b^	0.170
CIT	Citrulline	1.453	2.544	3.343	1.823	0.670
PHE	Phenylalanine	2.007 ^ab^	1.871 ^b^	2.589 ^a^	1.551 ^b^	0.241
TYR	Tyrosine	1.981 ^ab^	1.904 ^ab^	2.349 ^a^	1.781 ^b^	0.192
ASP	Aspartic acid	0.544 ^b^	0.871 ^ab^	0.972 ^a^	0.759 ^ab^	0.118
TRP	Tryptophan	0.633 ^a^	0.263 ^b^	0.332 ^ab^	0.152 ^b^	0.107
BAL	β-Alanine	0.398 ^a^	0.289 ^ab^	0.259 ^b^	0.326 ^ab^	0.044
ANS	Anserine	0.886 ^a^	0.218 ^b^	0.200 ^b^	0.084 ^b^	0.120
ASP	Asparagine	0.197	0.143	0.164	0.140	0.067

Data in the same row with different superscripts differ by a value of *p* < 0.05. SEM: standard error of the mean.

**Table 12 animals-15-02339-t012:** Nutritional and qualitative indexes of European sea bass fillets.

INDEX	25 ECO	30 ECO	35 ECO	30 CO	Reference Level	REFERENCE
Nutritional Indexes						
AI	0.18	0.19	0.20	0.22	<1	[11,72,76,89]
TI	0.00024	0.00029	0.00026	0.00038	<1	[11,72,76,89,90]
h/H	4.55	4.50	4.27	3.40	>1	[91]
FLQ	8.63	8.71	8.81	8.97	>1	[92]
Quality Indexes						
ω6/ω3	1.85	1.86	1.81	1.62	≤2	[92,93]
PUFA/SFA	2.06	2.04	1.90	1.60	>0.4	[93]
MUFA/SFA	1.31	1.27	1.25	1.31	>0.4	[93]
PUFA/MUFA	1.57	1.61	1.52	1.22	>0.67	[94]
(PUFA + MUFA)/SFA	3.36	3.31	3.15	2.92	>0.4	[93]
LA/ALA	6.51	6.61	6.84	7.37	>6.78–10.05	[95]

Atherogenic index (AI), thrombogenic index (TI), the ratio of hypocholesterolaemic over hypercholesterolaemic fatty acids (h/H), flesh lipid quality (FLQ), ω6/ω3 ratio, PUFA/SFA ratio, MUFA/SFA ratio, PUFA/MUFA ratio and (PUFA + MUFA)/SFA ratio. The recommended values (with references) are given.

**Table 13 animals-15-02339-t013:** Willingness to pay for sea bass obtained in the experimental trial, assessed using the Kruskal–Wallis test.

Sample	Frequency	Sum of Ranges	Ranges Rverage	Groups	
25 ECO	100	18,440.500	184.405	A	
30 CO	99	18,892.500	190.833	A	B
35 ECO	100	20,635.000	206.350	A	B
30 ECO	100	21,832.000	218.320		B

Results of the Kruskal–Wallis test at 95% confidence level.

**Table 14 animals-15-02339-t014:** Economic analysis of sea bass production fed with the different experimental feeds for 196 days.

	25 ECO	30 ECO	35 ECO	30CO	SEM
Diet Price (EUR kg^−1^)	1.69	1.66	1.62	1.29	
E.C.R. (EUR kg^−1^) ^1^	2.94 ^c^	2.59 ^b^	2.58 ^b^	1.91 ^a^	0.07
E.P.I. (EUR kg^−1^) ^2^	8.05 ^b^	8.41 ^a^	8.42 ^a^	3.58 ^c^	0.07

Means of triplicate groups. Values are presented as mean ± SEM (standard error of the pooled means). Values in the same row with different superscript letters are significantly different (*p* < 0.05). ^1^ Economic Conversion Ratio = FCR×Diet Price. ^2^ Economic Profit Index = Sea bass price − ECR (Considering a sell price of 5.5 EUR/kg for conventional sea bass and 11 EUR/kg for organic sea bass. The price of ingredients were: fishmeal (1601 EUR/ton), fish oil (3500 EUR/ton), organic wheat (499 EUR/ton), organic spelt bran (1950 EUR/ton), organic corn (300 EUR/ton), organic soymeal (1355 EUR/ton), organic soy oil (1815 EUR/ton), wheat (254 EUR/ton), soymeal (385 EUR/ton), soy oil (1145 EUR/ton), Calcium phosphate (3500 EUR/ton), taurine and methionine (2000 EUR/ton), vitamin complex (5000 EUR/ton).

## Data Availability

The data presented in this study are available on request from the corresponding author.

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
