# Peer review of "Effect of Organic Plant Ingredients on the Growth Performance of European Sea Bass (Dicentrarchus labrax): Nutritional Efficiency, Fillet Nutritional Indexes, Purchase Intention, and Economic Analysis"

_animals, 2025, doi:10.3390/ani15162339_

Round 1

Reviewer 1 Report

Comments and Suggestions for Authors

Dear authors,

Thank you for submitting your manuscript titled "Effect of plant organic ingredients on the growth of European sea bass (Dicentrarchus labrax)" to our journal. The research explores the impact of organic feed on sea bass, a topic of significant importance in the context of sustainable development in aquaculture. Upon careful review of your manuscript, I have identified several issues that require attention, which significantly affect the scientific rigor and readability of the manuscript. I recommend a major revision of the manuscript before considering resubmission.  

Specific issues:

  1. Consistency issue between research theme and experimental design:

The introduction and abstract of the manuscript indicate that the research objective is to explore the impact of "organic plant ingredients" on the growth of sea bass.  However, your experimental design primarily differentiates three organic feeds (25ECO, 30ECO, 35ECO) by adjusting the "fishmeal content" (25%, 30%, 35%), and compares them with a control feed containing 30% fishmeal of "conventional composition." This makes it difficult to directly assess the specific contribution of "organic plant protein" itself to growth and quality.  If the true objective of the study is to explore the substitutability of organic plant protein, it might be more straightforward to address this question by comparing the effects of organic versus conventional plant protein under a fixed fishmeal content (e.g., uniformly 30% fishmeal).  

Suggestion: Future experimental designs could consider adopting a two-factor design, for instance, one factor being fishmeal content (high, medium, low) and the other being plant protein source (organic, conventional), in order to more clearly isolate the impacts of different variables on the results.  Alternatively, if the research focus is on optimizing organic feed formulations, the rationale for using different fishmeal contents as internal variables within organic feeds should be explicitly stated.  

  1. Interpretation of growth performance results:

The manuscript indicates that the organic feed group did not demonstrate advantages in growth parameters, with the 25ECO group even showing negative impacts. This diverges from the expectation that organic aquaculture typically aims to provide sustainable and efficient alternatives.  The article mentions economic considerations, but if growth performance is suboptimal and organic raw materials are generally more costly, the market competitiveness requires more in-depth discussion and trade-offs.

  1. Title does not match content scope completely:

The paper's title focuses on "growth," but the actual content encompasses multiple dimensions, including the nutritional composition of fish meat, amino acid profiles, fatty acid profiles, consumer preferences, and economic assessments. While this demonstrates the comprehensiveness of the research, it also results in a lack of focus in the article.

  1. Language quality and readability:

 The manuscript exhibits widespread issues in English expression, including but not limited to grammatical errors, inaccurate word usage, chaotic sentence structure, and lack of fluency. This severely affects the reading experience and professional impression.  For example:

"To Free amino acids profle, fat was extracted…" This sentence is both grammatically and logically incorrect.  Fat extraction is typically used for fatty acid analysis, rather than free amino acid analysis.  

In the method description, the term "interphase" was inappropriately used, as the supernatant or pellet is usually collected after centrifugation.  

The chemical measurement units "ml" and "μl" should be standardized to "mL" and "μL", respectively.  

Although the research topic is valuable, the current manuscript exhibits significant deficiencies in experimental design rigor, clarity of results presentation, and language quality.  I recommend that you resubmit after making substantial revisions. In particular, it is necessary to re-examine whether the experimental design can truly address the core questions you intend to investigate, and to conduct a thorough review of the language and scientific rigor throughout the manuscript. Thank you for your efforts and contributions to academia.

Author Response

Response 1: [Firstly, the objective is to optimize organic feed with a reduced inclusion of fishmeal for sustainable organic aquaculture. Despite the absence of synthetic amino acids and the lack of certain raw materials of plant or terrestrial livestock origin, the study's findings are noteworthy. The analysis of the quality of the fillet, its variation according to its profile of fatty acids and free amino acids, and the nutritional indices of the fillets, as determined by Sensometrics studies, are also valuable insights. In addition, we will assess consumer perceptions and purchase intentions for the treatments, along with an economic analysis. While other studies on organic aquaculture have highlighted high costs due to the use of fishmeal (50-60%), our study focuses on lower levels of inclusion.

4. Response to Comments on the Quality of English Language

Point 1:  recommend that you resubmit after making substantial revisions

Response 1: .The description of the analysis methodologies for amino acids, fatty acids, free amino acids, and fatty acid profile of the fillet, among others, has been corrected.

The title of the article has also been improved, justifying the analyses carried out, as well as the well-defined objective and the final conclusions reached.  

5. Additional clarifications

[This research starts from the principle of optimizing organic feed with a lower inclusion of fishmeal, for organic and sustainable aquaculture, despite the fact that no synthetic amino acids are added and that some raw materials of both plant and terrestrial livestock origin lack them. The sea bass fillets in this study were subjected to Sensometrics studies in CATA tests by untrained panelists, with their evaluation of purchase intention among the treatments evaluated being important, followed by an economic analysis. There is other research carried out on other species in organic aquaculture, but they are not as sustainable due to the inclusion of 50 to 60% fishmeal in their formulations. whereas working with by-products from livestock and organic aquaculture can reduce the impact generated by the use of fishmeal, but the fillets are affected both in their lipid profile and in the presence of free amino acids, which were affected by the level of fishmeal inclusion when the CATA studies were carried out. However, the nutritional indices of the fillets in this study were within the established references, but still with very low values that could be improved by increasing those values (TI, AI, h/H, FLQ, etc.) and optimising organic feed with new sustainable raw materials.

Reviewer 2 Report

Comments and Suggestions for Authors

This manuscript provides original data on the Effect of plant organic ingredients on the growth of European sea bass (Dicentrarchus labrax).

The MS is generally well written with some appropriate data analyses and interesting discussion. The topic is original and relevant to the field.

All references and figures are appropriate. Nevertheless, it has a number of shortcomings which should be addressed by the authors in order to improve understanding of the ms.

Line 67, Change ‘the’ to The

Was there any feed that was not consumed by fish?  Please write line 138.

Line 162, How much vitamin does the premix contain, write the unit as g kg-1 or mg kg-1.Line 162, Vitamin and mineral complex does not contain minerals,

Line 162, Change the ‘ascorbicacid’ to ascorbic acid

Line 162, 166, doble ascorbic acid

Line 162, 164, double DL-a-tocopherol

Line 163, (PO4)2Ca3 change to (PO4)2Ca3

Line 164, What is rivoflamine? Is it riboflavine (vitamin B2)? 

Line 182, table 2 to Table 2

Line 219, Formula writing should be simplified

Line 363, Table 5, Superscripts on the FI and FCR, why?

Line 378, Table 6, Biometric parameters (CF, VSI, HSI and MF) contains superscript letters, why?

Line 815, Penaeus monodon must be italic.

Line 843, Cyprinus carpio  must be italic.

Line 867, Paralichthys olivaceus must be italic.

All the species names should be italic  throughout the text.

Comments on the Quality of English Language

This manuscript provides original data on the Effect of plant organic ingredients on the growth of European sea bass (Dicentrarchus labrax).

The MS is generally well written with some appropriate data analyses and interesting discussion. The topic is original and relevant to the field.

All references and figures are appropriate. Nevertheless, it has a number of shortcomings which should be addressed by the authors in order to improve understanding of the ms.

Line 67, Change ‘the’ to The

Was there any feed that was not consumed by fish?  Please write line 138.

Line 162, How much vitamin does the premix contain, write the unit as g kg-1 or mg kg-1.Line 162, Vitamin and mineral complex does not contain minerals,

Line 162, Change the ‘ascorbicacid’ to ascorbic acid

Line 162, 166, doble ascorbic acid

Line 162, 164, double DL-a-tocopherol

Line 163, (PO4)2Ca3 change to (PO4)2Ca3

Line 164, What is rivoflamine? Is it riboflavine (vitamin B2)? 

Line 182, table 2 to Table 2

Line 219, Formula writing should be simplified

Line 363, Table 5, Superscripts on the FI and FCR, why?

Line 378, Table 6, Biometric parameters (CF, VSI, HSI and MF) contains superscript letters, why?

Line 815, Penaeus monodon must be italic.

Line 843, Cyprinus carpio  must be italic.

Line 867, Paralichthys olivaceus must be italic.

All the species names should be italic  throughout the text.

Author Response

Comments 1: reviewer 2

Response 1: [Thank you very much for your observation.

Line 67 has indeed been corrected from ‘the’ to ‘The’ and is now line 69.

All fish consumed the formulated feed, even the feed with a low level of fishmeal inclusion (25 ECO).

Line 167, previously 162, the vitamin is expressed in g kg -1 and also contains a premix, which is detailed below, and the observations in the text have been corrected.

Line 182, now 187, was changed from Table 2 to Table 2.

Line 363, now 367, was corrected from the superscripts.

Line 378, now 382, was corrected from the superscripts and their interpretation as it should be.

Line 815, now 812, has been corrected in italics to Penaeus monodon.

Line 867, Paralichthys olivaceus, has been corrected in italics,

and all scientific names of species cited in the bibliographical references have been corrected in italics.

4. Response to Comments on the Quality of English Language

Point 1:

Response 1: (The description of the analysis methodologies for amino acids, fatty acids, free amino acids, and fatty acid profile of the fillet, among others, has been corrected.

The title of the article has also been improved, justifying the analyses carried out, as well as the well-defined objective and the final conclusions reached)

5. Additional clarifications

[This research starts from the principle of optimizing organic feed with a lower inclusion of fishmeal, for organic and sustainable aquaculture, despite the fact that no synthetic amino acids are added and that some raw materials of both plant and terrestrial livestock origin lack them. The sea bass fillets in this study were subjected to Sensometrics studies in CATA tests by untrained panelists, with their evaluation of purchase intention among the treatments evaluated being important, followed by an economic analysis. There is other research carried out on other species in organic aquaculture, but they are not as sustainable due to the inclusion of 50 to 60% fishmeal in their formulations.

Round 2

Reviewer 1 Report

Comments and Suggestions for Authors

The overall paper has undergone significant revisions. But some small details need to be paid attention to, such as the standardization of international units (such as line 260), and references can be appropriately simplified, citing classic and more recent ones (such as line 118).

Author Response

Thank you very much for your comment. Please find attached our response.

Reviewer 2 Report

Comments and Suggestions for Authors

The manuscript has been sufficiently improved to warrant publication in Animals. The necessary corrections have been made and are shown in the text. English editing has also been done.

Line 67, Change ‘the’ to The- OK

Was there any feed that was not consumed by fish?  Please write line 138.- OK

Line 162, How much vitamin does the premix contain, write the unit as g kg-1 or mg kg-1.-OK

Line 162, Vitamin and mineral complex does not contain minerals,-OK

Line 162, Change the ‘ascorbicacid’ to ascorbic acid- OK

Line 162, 166, doble ascorbic acid-OK

Line 162, 164, double DL-a-tocopherol-OK

Line 163, (PO4)2Ca3 change to (PO4)2Ca3- OK

Line 164, What is rivoflamine? Is it riboflavine (vitamin B2)? Spanish and English words are mix- OK

Line 182, table 2 to Table 2-OK

Line 219, Formula writing should be simplified-OK

Line 363, Table 5, Superscripts on the FI and FCR, why?-OK

Line 378, Table 6, Biometric parameters (CF, VSI, HSI and MF) contains superscript letters, why?- OK

Line 815, Penaeus monodon must be italic.- OK

Line 843, Cyprinus carpio  must be italic.- OK

Line 867, Paralichthys olivaceus must be italic.- OK

All the species names should be italic throughout the text.-OK

Author Response

(The authors gave the same response as above.)
